# Results and Clinical Interpretation of Germline *RET* Analysis in a Series of Patients with Medullary Thyroid Carcinoma: The Challenge of the Variants of Uncertain Significance

**DOI:** 10.3390/cancers12113268

**Published:** 2020-11-05

**Authors:** Giovanni Innella, Cesare Rossi, Maria Romagnoli, Andrea Repaci, Davide Bianchi, Maria Elena Cantarini, Davide Martorana, Lea Godino, Andrea Pession, Antonio Percesepe, Uberto Pagotto, Daniela Turchetti

**Affiliations:** 1Division of Medical Genetics, Azienda Ospedaliero-Universitaria di Bologna, 40138 Bologna, Italy; giovanni.innella@studio.unibo.it (G.I.); cesare.rossi@unibo.it (C.R.); maria.romagnoli@uslcentro.toscana.it (M.R.); lea.godino@aosp.bo.it (L.G.); 2Department of Medical and Surgical Sciences, University of Bologna, 40138 Bologna, Italy; andrea.pession@unibo.it (A.P.); uberto.pagotto@unibo.it (U.P.); 3Endocrinology and Diabetes Prevention and Care Unit, Department of Medical and Surgical Sciences, University of Bologna, 40138 Bologna, Italy; rep.rep@libero.it; 4Division of Endocrinology, Ospedale di Bentivoglio, 40010 Bentivoglio (BO), Italy; davide.bianchi@ausl.bo.it; 5Division of Pediatric Oncology, Azienda Ospedaliero-Universitaria di Bologna, 40138 Bologna, Italy; mariaelena.cantarini@aosp.bo.it; 6Division of Medical Genetics, Azienda Ospedaliero-Universitaria di Parma, 43126 Parma, Italy; dmartorana@ao.pr.it (D.M.); antonio.percesepe@unipr.it (A.P.)

**Keywords:** *RET*, medullary thyroid carcinoma, clinical management, variants of uncertain significance

## Abstract

**Simple Summary:**

About 25% of Thyroid Carcinomas of Medullary type occur in carriers of hereditary alterations in the *RET* gene. Different alterations are associated with different risks (highest, high, and moderate) and management depends on risk category. We explored prevalence, clinical presentation and management of inherited *RET* variants in patients tested at our center. We found inherited *RET* variants in 31.9% of tested individuals: the vast majority of patients with Medullary Thyroid Carcinoma who had a family history positive for the disease was found to carry a *RET* change, but also 14.3% of those with no family history tested positive, supporting the recommendation to perform genetic testing in all cases of Medullary Thyroid Carcinoma. For known variants, findings in our patients were consistent with available risk classification. Besides, we obtained evidence supporting the classification of two rare variants of uncertain clinical significance (p.Ser904Phe and p.Asp631_Leu633delinsGlu), which may help future management of carriers.

**Abstract:**

Germline *RET* variants are responsible for approximately 25% of medullary thyroid carcinoma (MTC) cases. Identification of *RET* variant carriers allows for the adoption of preventative measures which are dependent on the risk associated with the specific alteration. From 2002 to 2020, at our cancer genetics clinic, *RET* genetic testing was performed in 163 subjects (102 complete gene analyses and 61 targeted analyses), 72 of whom presented with MTC. A germline *RET* variant was identified in 31.9% of patients affected by MTC (93.8% of those having positive family history and 14.3% of clinically sporadic cases). Subsequent target testing in relatives allowed us to identify 22 asymptomatic carriers, who could undertake appropriate screening. Overall, patients with germline *RET* variants differed significantly from those who tested negative by family history (*p* < 0.001) and mean age at MTC diagnosis (44.45 vs. 56.42 years; *p* = 0.010), but the difference was not significant when only carriers of moderate risk variants were considered (51.78 vs. 56.42 years; *p* = 0.281). Out of 12 different variants detected in 49 patients, five (41.7%) were of uncertain significance (VUS). For two of these, p.Ser904Phe and p.Asp631_Leu633delinsGlu, co-segregation and genotype/phenotype analysis, matched with data from the literature, provided evidence supporting their classification in the moderate and the highest/high risk class (with a MEN2B phenotype), respectively.

## 1. Introduction

Medullary thyroid carcinoma (MTC) accounts for 4–10% of all thyroid cancer cases [1,2] and originates from the parafollicular cells of the thyroid or C-cells, physiologically responsible for the secretion of calcitonin; C-cell hyperplasia (CCH) is considered as the first stage of histological progression that evolves into MTC [3,4]. Among MTC cases, around 25% present in the context of an inherited syndrome, whereas the remaining 75% are sporadic [5,6,7]. Inherited MTC syndromes include multiple endocrine neoplasia type 2 (MEN2A and MEN2B) and familial MTC (FMTC) [8,9]. MTC is associated with pheocromocytoma (PCC) and parathyroid hyperplasia/primary hyperparathyroidism (PHPT) in MEN2A, [10,11], and with PCC, marfanoid habitus and ganglioneuromatosis of the gut and oral mucosa in MEN2B [12]; conversely, FMTC is characterized by multiple cases of MTC in the family, with no other clinical manifestations [13].

Both MEN2 and FMTC are caused by germline, activating variants in the *RET* proto-oncogene following an autosomal dominant pattern of transmission [14,15]. Nevertheless, different *RET* mutations confer different cancer risks [16,17,18,19], which has led to recommendations for management of the carriers based on the level of risk associated with the specific mutation [20]. In this regard, the American Thyroid Association (ATA) stratified *RET* mutations into distinct risk levels, in order to define the most appropriate management for each known mutation. Former classification provided for a stratification into four risk levels (from D to A from the highest to the lowest level of risk) [21], while the 2015 guideline revision combined levels A and B for a total of three risk levels: highest (HST), high (H) and moderate (MOD) [22].

The ATA’s recommended management consists of prophylactic thyroidectomy to be performed as soon as possible in carriers of HST and H risk level variants (within the first year of life and within 5 years, respectively), while for carriers of MOD risk level variants, it is suggested to perform annual serum calcitonin screening, and prophylactic thyroidectomy should be performed when values become elevated. Screening for PCC (annual dosing of plasma and urinary catecholamines and metabolites) should start at 11 years for carriers of H/HST variants and at 16 years for carriers of MOD variants. For carriers of H and MOD variants, biochemical screening for PHPT is also recommended (annual dosing of serum calcium and parathyroid hormone), starting at 8 and 20 years, respectively [21,22,23].

Of course, a stratification by risk is possible only for known, recurrent mutations, for which genotype–phenotype correlations have been clearly established. Conversely, the detection of a novel or rare mutation in a family poses serious counselling and management issues, as the associated risk is mostly unknown. In these cases, until genotype–phenotype correlations are clarified, risk and subsequent management of healthy carrier cannot be based on standard guidelines and need to be evaluated case-by-case, possibly on the basis of clinical and family history of the carriers, which however, may be informative only for large families with many well characterized members [24].

The aim of the study was to analyze the experience of our center with *RET* genetic testing in order to (a) explore prevalence, clinical significance and genotype–phenotype correlations of germline *RET* mutations, (b) collect information which could potentially contribute to defining the clinical role of the variants of uncertain significance (VUS) encountered at our center, and (c) discuss the approach to management of carriers of novel and rare *RET* mutations.

## 2. Results

### 2.1. Clinical Characteristics and Genetic Test Results

Among subjects submitted to *RET* analysis in our laboratory, 117/163 (71.8%) had personal and/or family history of MTC (114) or CCH (3), while the remaining had personal and/or family history of other pathologies possibly related to *RET* alterations (PCC, PHPT, intestinal ganglioneuromatosis). The list of cases included in the study is reported in Appendix A.

A germline *RET* variant was identified in 49/163 (30.1%) subjects: 15/102 (14.7%) index cases, who underwent a complete gene analysis, and 34/61 (55.7%) relatives of a *RET* carrier, who underwent a targeted search for a known variant. Considering only the 72 patients affected by MTC/CCH, a germline *RET* variant was identified in 23 (31.9%), with 15/16 (93.8%) of these having positive family history and 8/56 (14.3%) of sporadic cases. Finally, considering only patients with CCH, 1/3 (33.3%) carried a variant of *RET*.

Mean age at MTC diagnosis was 44.45 years in patients with germline *RET* variants (pathogenetic or considered likely pathogenetic) and 56.42 years in patients with no variants detected (*p* = 0.010). Among patients with known cancer staging, 42.9% of those with *RET* variants and 36.4% of those with negative analysis had locally advanced disease (T > 1). All four MTC patients who also displayed other manifestations of MEN2 were found to carry a pathogenic germline *RET* variant.

### 2.2. RET Test Results and Clinical Correlations

Twelve different *RET* variants were identified in our sample: seven previously classified in a specific risk class (one of HST risk level, two of H risk level, four of MOD risk level) (Table 1) and five unclassified. The HST risk level variant p.Met918Thr was identified in a sporadic case of MEN2B, as well as in an external patient for whom phenotype was not detailed. H risk level variants were identified in two cases of familial MEN2A. MOD risk level variants were found in 34 subjects of 13 families; all belonged to A risk class (the lowest risk level) in the previous classification.

Overall, patients with germline *RET* variants significantly differed from those testing negative by the presence of family history (68.2% vs. 2.1%; *p* < 0.001) and by mean age at MTC diagnosis (44.45 vs. 56.42; *p* = 0.010). When comparing carriers of MOD risk variants with wild-type *RET* patients, the presence of family history was still significantly more frequent in the former group (72.2% vs. 2.1%; *p* < 0.001), while mean age at MTC diagnosis did not differ significantly (51.78 vs. 56.42; *p* = 0.281). Indeed, mean age at MTC diagnosis was 51.78 years among carriers of MOD risk variants, compared to 11.5 years in carriers of HST/H risk level variants (*p* < 0.001). The comparison of clinical features of patients with MTC according to test results is shown in Table 2.

### 2.3. Families with Unclassified Variants

Twelve individuals were carriers of variants for which the associated risk has not been clearly established. Overall, five VUS were found: one in multiple individuals from a single family, while the others were identified in individual subjects. Details on the variants are shown in Table 3.

The p.Ser904Phe variant was found in a family with several members affected by MTC (Figure 1): eight individuals were tested and found to carry the variant, while two other relatives were obligate carriers. Out of 10 carriers, seven developed slowly progressing MTC at an average age of 46.3 years and none manifested other *RET*-related problems. p.Ser904Phe is a rare variant previously reported in one family with father and son affected by adult-onset MTC [25]; therefore, the associated risk is still unclear. Cosci et al. [26], through in silico and in vitro analyses, showed that the variant has relatively high transforming activity but low aggressiveness and suggested to assign the variant to the lowest ATA risk level A. Consistently, the segregation of the variant in our family and the clinical history of carriers showed that, although highly penetrant, this variant causes late-onset, slowly progressing MTC, leading us to hypothesize that the screening recommended for carriers of lowest-risk mutations may be appropriate for healthy carriers of the p.Ser904Phe variant.

The p.Tyr791Phe variant was found in a patient with MTC diagnosed at 22 years of age and a negative family history for endocrine diseases. The variant, first reported in patients with Hirschsprung disease [27], MTC [28,29] and PCC [30], involves a highly conserved amino acid and used to be regarded as pathogenic based on in silico predictions. More recently, the evidence that the variant has similar frequencies in affected and unaffected subjects [31,32] is more common in the population than expected for a disease-causing variant [33,34], fails to co-segregate with the disease in some families [35,36] and co-occurred with a pathogenic variant in some patients [37,38] led researchers to reconsider it as likely benign. In our patient, however, the young age at MTC diagnosis raises the suspicion that this variant may have, to some extent, favored the development of MTC, possibly interacting with other factors in a multifactorial context, or that she carries a pathogenic variant undetected by the multigene test performed.

The p.Lys710Arg variant was found in a single 76-year-old patient who had PHPT and elevated serum calcitonin, who underwent a multigene analysis for hyperparathyroidism. This variant, very rare in population databases, has never been reported in patients affected by conditions known to be *RET*-related. The lysine residue substituted by arginine at codon 710 of the protein is highly conserved, but there is a small physicochemical difference between the two amino acids; consequently, computational predictors give conflicting results on the potential impact of this missense change, which is currently classified as of uncertain significance. As the clinical picture of our patient is not strongly suggestive of a *RET*-related condition, it is likely that this variant did not play a significant role in his disorder.

The p.Ser649Leu variant was found in a patient with MTC diagnosed at 59 years of age and a negative family history for endocrine diseases who also carried the common p.Val804Met variant, belonging to the MOD risk level. Her unaffected daughter was found to carry only the p.Val804Met variant, demonstrating that the two variants are “in trans” in the proband. The variant is rare in population databases and multiple lines of computational evidence reported by Varsome support its deleterious effect on the gene or gene product, but conflicting interpretations regarding its pathogenicity are present in the literature [36,39]. The evidence that, in our patient, the p.Ser649Leu variant is present “in trans” with a known pathogenic variant is against its pathogenicity, although an additive effect in combination with p.Val804Met cannot be excluded.

The p.Asp631_Leu633delinsGlu (c.1893_1898delCGAGCT) variant causes an in-frame deletion of two amino acids; the nomenclature reflects the fact that the deletion starts at the third base of the Asp631 codon and extends through Glu632 up to the second base of Leu633, resulting in deletion of Glu632 and Leu633 and a change of the Asp631 codon into glutamic acid. The deletion was identified in a female child with neonatal onset of abdominal distension, constipation and vomiting, with subsequent growth retardation, who was diagnosed with abdominal-pelvic plexiform ganglioneuroma when she was 2 years old. This patient also displayed lesions consistent with neurofibromas at the buttocks, mild dysmorphic features and nodules at the upper lip and was diagnosed with MTC and parathyroid adenoma when she was 7 years old. This clinical picture, resembling the MEN2B phenotype, led us to perform *RET* testing. The p.Asp631_Leu633delinsGlu variant is not reported in population databases, or in the medical literature; therefore, its clinical impact was completely unknown. Segregation analysis undertaken in the family demonstrated that it occurred “de novo”, supporting its pathogenicity. Generally, *RET* defects associated with MEN2 and FMTC are typically gain of function, while deletions of part of the gene are mostly expected to cause loss of function; however, cases of *RET* deletions associated with MEN2 have been reported [40]. Moreover, Borganzone et al. studied a similar somatic alteration of *RET*, p.Glu632_Leu633del (c.1894_1899delGAGCTG), and demonstrated that this in-frame deletion reduces the spacing between two Cysteine residues, causing ligand-independent constitutive dimerization and activation of RET. Remarkably, RET activation was even greater in this case compared to activation induced by the frequent mutation p.Cys634Arg [41,42]. Although in our variant the deletion is shifted by 1 bp compared to the somatic mutation described by Borganzone et al., it results in the deletion of two amino acids in the same location; thus, the final effect—that is, the constitutive activation of RET signaling—is likely to be the same. Collectively taken, the “de novo” origin in our patient, her clinical phenotype and the functional data support the hypothesis that the p.Asp631_Leu633delinsGlu variant is causative of MEN2B and should be assigned to the HST/H risk class.

## 3. Discussion

Since the important role of *RET* alterations in the pathogenesis of MTC has been ascertained, it is considered appropriate to perform the analysis of this gene in all individuals diagnosed with primary CCH, MTC or MEN2, either sporadic or familial [3]. The identification of positive subjects is in fact important for the management of patients and especially of their healthy relatives, who could benefit from specific surveillance programs and/or prophylactic treatments [43,44]. The aim of this study was to critically analyze our experience with *RET* testing.

Among the 72 individuals affected by MTC or CCH who underwent *RET* analysis at our laboratory between 2002 and 2020, 23 (31.9%) were found to carry a germline *RET* variant. As expected, a positive family history increased the chance of finding a variant: out of 16 subjects who had positive family history, 15 had a detectable *RET* variant. In the only familial case where the analysis was negative (subject 51), the patient had been treated at 32 years of age for primary CCH and her father had died at 43 years of age for MTC; given the strong suspicion of an underlying genetic cause, a second-level analysis was performed using the NGS multi-gene panel for endocrine tumors, but no variants were found. Despite the negative test results, the early age of onset and the family history are suspicious for an undetected MTC-predisposing gene defect.

Germline *RET* variants were identified in eight of 56 subjects with apparently sporadic MTC (14.3%, slightly higher than the 4–10% reported in the literature [26,45,46,47]), confirming the importance of screening *RET* in all cases of MTC, even when the family history is negative [48]. Indeed, testing allowed us to reclassify as hereditary a fraction of apparently sporadic cases and led us to extend genetic testing to 18 relatives, eight of whom (44.4%), belonging to three distinct families, were found to carry the variant (p.Val804Met in all cases) and were therefore able to benefit from specific surveillance.

Among patients with CCH, 1/3 (33.3%) carried a *RET* variant (of MOD risk level); actually, she was an asymptomatic patient who was found to have inherited the familial variant and subsequently undertook the surveillance program that led to the diagnosis of CCH. On one hand, this may have represented a successful instance of early diagnosis, provided that CCH is a precursor condition for MTC; on the other hand, we cannot be sure that this benign finding would eventually evolve into a malignant condition and this may have been, conversely, a case of overdiagnosis and overtreatment.

Taking into account clinical features and genetic test results shows that wild-type *RET* patients and carriers of MOD risk level variants only differ by family history (*p* < 0.001), but not by clinical characteristics such as mean age at diagnosis of MTC and cancer stage. This is also supported by the fact that among 40 individuals found to carry variants assigned to MOD risk level or the p.Ser904Phe variant (excluding those for whom we have no clinical information), 22 had not developed primary CCH or MTC at the time of the last follow-up (55.0%): 4/9 of those over 60 years of age (44.4%), 8/18 of those between 40 and 60 years of age (44.4%), 8/11 of those between 20 and 40 years of age (72.7%) and 2/2 of those under 20 years of age (100.0%). Several lines of evidence have suggested that the aggressiveness of MTC does not depend on the presence, absence or type of *RET* variant (which mainly affects the age at onset of the disease) but on the stage and the age at diagnosis of the disease [20], which are the strongest predictors of survival for patients with MTC; therefore, our data further support the appropriateness of the non-invasive screening recommended by current guidelines for healthy carriers of MOD risk level variants.

For carriers of variants with unclear associated risk, however, until the genotype–phenotype correlations are clarified, the attempt to assess the risk and respective proper management of healthy carriers only relies on the clinical history of carriers—which, however, can provide meaningful information only when large families with many characterized members are available—or on any significant biomolecular evidence.

Thus, in the 91-O-03 family, the availability of several genetically and clinically characterized members allowed us to provide evidence that the p.Ser904Phe variant is highly penetrant (7/10 of carriers developed MTC, 70%) but leads to the development of slowly-progressing MTC at relatively advanced age (average age at diagnosis: 46.3 years), suggesting that recommended screening for carriers of lower risk mutations is appropriate for healthy carriers of this variant.

Moreover, in the case of the p.Asp631_Leu633delinsGlu variant, the clinical picture of the patient and the “de novo” origin, associated with the functional studies on a very similar variant [41,42], provide convincing evidence in favor of the pathogenicity of p.Asp631_Leu633delinsGlu and of its assignment to the HST/H risk level category. Of note, clinical manifestations in this patient were consistent with MEN2B, a phenotype that has been reported to be associated in 95% of cases with the p.Met918Thr variant and in 2–3% of cases with the p.Ala883Phe variant [49]. Both these variants affect residues located in the substrate specificity pocket of the central catalytic core of the tyrosine kinase domain and likely cause RET activation by altering its substrate specificity [50]. Conversely, p.Asp631_Leu633delinsGlu affects residues located at a great distance in a different domain (the cysteine-rich extracellular domain). Mutations in this domain are expected to cause RET activation by inducing its disulfide-linked dimerization and are generally associated with a MEN2A/FMTC phenotype, which gives a new perspective in the view of elucidating molecular mechanisms leading to the more severe MEN2B phenotype. Rarely, a MEN2B-like phenotype has been described in patients carrying two *RET* variants (bi-allelic or *in-cis* on the same allele) [51,52]. It can be hypothesized that, although different, all these defects result in a particularly intense RET activation and that the higher the activation level, the more severe is the phenotype.

Among the families with MOD risk level variants, the case of the 228-O-18 family, carrying the p.Leu790Phe variant, is worthy of consideration. In this family, the p.Leu790Phe variant was found in nine individuals, four of whom developed MTC; intriguingly, three of these individuals were also affected by neurofibromatosis 1 (NF1), caused by a mutation of the *NF1* gene inherited from the other parental branch, and one developed a PCC. The p.Leu790Phe variant is classified at the lowest risk level and is generally associated exclusively with MTC [53]; it is therefore possible that, in this individual, the risk of developing a PCC was greater than that of the ordinary carriers of this variant, due to the co-presence of the mutation of *NF1*, another gene whose alterations are associated with an increased risk of developing PCC.

One limitation of this study is that most patients underwent the analysis of selected *RET* exons through Sanger sequencing, which is expected to be less sensitive if compared to whole-gene NGS-based analysis. However, since all the variants identified were found using the Sanger method, and the percentage of individuals is in line with data previously reported in the literature (even slightly higher for sporadic cases), we can conclude that this testing approach demonstrated satisfactory accuracy in finding *RET* variants, supporting the evidence that most clinically relevant variants reside in known mutational hotspots [54].

## 4. Materials and Methods

### 4.1. Patients

From 2002 to 2020, in our laboratory, *RET* molecular analysis was performed in a total of 163 subjects, 102 of which underwent a complete gene analysis, while in the other 61, the targeted search for a family mutation was performed. In total, 120 of these subjects had participated in genetic counseling at our Cancer Genetics Clinic in Bologna based on personal history of a possible *RET*-related condition or identification of a *RET* mutation in the family; after verifying the presence of criteria for *RET* testing, informed consent was collected and a venous blood sample was drawn. For the other 43 subjects, blood sample was sent to our laboratory by external centers with the request for *RET* analysis, after informed consent had been collected by the requesting physician.

### 4.2. Clinical Data

The phenotype leading to the suspicion of a *RET* variant in the family, and therefore the reason for *RET* analysis, was known for 140 of the 163 analyzed subjects. Of these, main clinical information regarding families with MTC, including disease status, age at diagnosis and family history, was available for all the patients who came to our clinic for genetic counseling and for the other 15 subjects sent from external physicians. This and any other information, such as stage of MTC, survival status and presence of any further pathologies, was collected during genetic counseling and/or derived from medical records and pathology reports.

### 4.3. RET Analysis

Genomic DNA was isolated from peripheral blood-EDTA using the QIAmp DNA Blood Mini Kit according the manufacturer’s protocol (Qiagen, Valencia, CA, USA).

From 2002 to June 2019, sequencing of exons 5, 8, 10, 11, 13, 14, 15 and 16 of *RET* (RefSeq.NM_020975.5) was performed through bidirectional Sanger sequencing: briefly, PCR amplifications of target exons were carried out with FastStartTaq DNA polymerase (Roche, Basel, Switzerland), followed by standard dideoxy sequencing, and run on a ABI3730 DNA analyzer (Applied Biosystems, Foster City, CA, USA). PCR and sequencing conditions, as well as the primer sequences, are available upon request. Chromatograms were analyzed for variants using the software Sequencer (Gene Code Corporation, Ann Arbor, MI, USA). According to the ARUP database as of January 2020 (https://arup.utah.edu/database/MEN2/MEN2_display.php) exons 5, 8, 10, 11, 13, 14, 15 and 16 cover all but one (Exon 7: c.1513_1518delGAGGGG: p.E505_G506del) of the *RET* mutations described in the literature.

Since July 2019, the entire RET cds (20 exons) has been included in an NGS panel (IAD177392-ThermoFisher Scientific) comprising 27 genes related to MEN2, pheochromocytoma and renal carcinoma and run on an Ion S5 next-generation sequencing system followed by analysis with the Ion Reporter Software (ThermoFisher Scientific, Waltham, MA, USA). The mean amplicon coverage and the target base coverage at > 100x for multiple experiments were > 800x and > 97%, respectively: these parameters guarantee a sensitivity of > 99% for the test. All the variants of interest were confirmed through Sanger sequencing.

Overall, 157 patients underwent Sanger sequencing analysis and 6 the NGS panel analysis.

### 4.4. Interpretation of Unclassified Variants

Rare/novel variants whose associated risk was unknown were evaluated through a review of the information available in the following public databases: gnomAD, ClinVar, varsome. In particular, for each variant, the classification in the mutational databases, the frequency in the population databases, the conservation of the substituted amino acids, the results of in silico predictions about the effect of the variant on the protein and the results of any functional assay were evaluated. All databases were last consulted on 15 September 2020. Moreover, reports about these variants possibly present in the literature were researched and evaluated. Finally, when possible, segregation of the variant in the family was assessed, in particular for variants classified as probably pathogenic and if/when other cases of MTC were present in the family.

### 4.5. Statistical Analysis

All available data were entered anonymously into a dedicated database and were analyzed by using the statistical package IBM-SPSS Statistics (Ver. 25 for Windows, IBM Co., Armonk, NY, USA). Means, standard deviation (SD), ranges and frequencies were used as descriptive statistics. The Fisher’s exact test was used for dichotomous variables and the independent t-test to analyze differences between two group means. Two-tailed *p* values lower than 0.05 were considered statistically significant. For the analysis, carriers of the p.Ser904Phe variant were included in the MOD category and the carrier of p.Asp631_Leu633delinsGlu variant in the HST/H category. Carriers of the other VUS were excluded from the analysis.

## 5. Conclusions

The results of our study provide support to the recommendation that *RET* genetic screening should be performed in all MTC cases, regardless of the family history of patients and their clinical presentation, and according to the appropriateness of ATA’s guidelines for clinical management of carriers of MOD risk level mutations. It is also highlighted that *RET* molecular analysis leads to the detection of a substantial proportion of variants associated with unknown risks, which poses serious challenges to the counselling and management of the patients and the family. However, co-segregation analysis in the family, genotype/phenotype analysis and a careful revision of the databases and literature proves helpful, at least in some of the cases, in order to tentatively assign the case to one of the known risk classes and inform management accordingly. This approach led us to provide evidence supporting the classification of p.Ser904Phe as the lowest risk level variant and of p.Asp631_Leu633delinsGlu as a novel variant responsible for MEN2B of HST/H risk level.

## Figures and Tables

**Figure 1 cancers-12-03268-f001:**
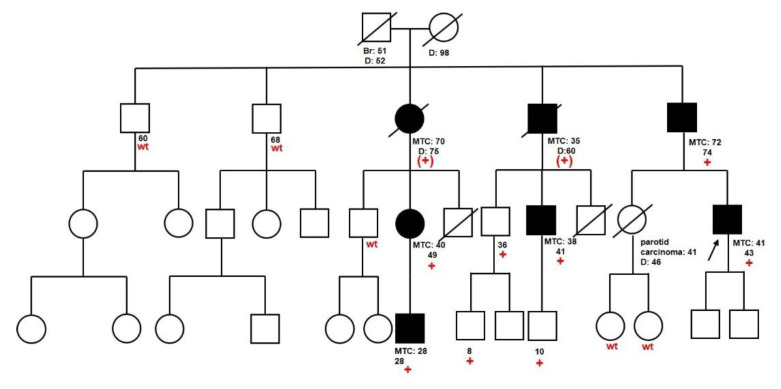
Family tree of family 91-O-03. We report the age at diagnosis of MTC and the age at death or at the last follow-up of patients evaluated at our clinic and/or who developed MTC; in red, we indicate the result of *RET* analysis (“+” = carrier of p.Ser904Phe variant; “(+)” = obligate carrier of the variant; “wt” = testing negative).

**Table 1 cancers-12-03268-t001:** *RET* classified variants detected in the population under study.

Nucleotide Variant	Aminoacidic Change	ATA Risk Level	N. Families	N. Carriers	MTC/tot Carriers ^a^ (%)	PCC/Tot Carriers ^a^ (%)	PHPT/Tot Carriers ^a^ (%)	MEN2B Manifestations ^b^/Tot Carriers ^a^ (%)
c.2671T > G	p.Ser891Ala	MOD	1	1	0/1 (0.0)	0/1 (0.0)	0/1 (0.0)	0/1 (0.0)
c.2410G > A	p.Val804Met	MOD	5	14	6/12 (50.0)	0/12 (0.0)	0/12 (0.0)	0/12 (0.0)
c.2370G > T	p.Leu790Phe	MOD	1	9	7/9 (77.8)	1/9 (11.1)	0/9 (0.0)	0/9 (0.0)
c. 2304G > C	p.Glu768Asp	MOD	2	10	3/10 (30.0)	0/10 (0.0)	0/10 (0.0)	0/10 (0.0)
c.1901G > T	p.Cys634Phe	H	1	1	1/1 (100.0)	1/1 (100.0)	0/1 (0.0)	0/1 (0.0)
c.1901G > A	p.Cys634Tyr	H	0	1	1/1 (100.0)	1/1 (100.0)	0/1 (0.0)	0/1 (0.0)
c.2753T > C	p.Met918Thr	HST	2	2	2/2 (100.0)	0/1 (0.0)	0/1 (0.0)	1/1 (100.0)

^a^ Only carriers with known clinical information were included; ^b^ Including marfanoid habitus, ganglioneuromatosis of the gut and oral mucosa, mild dysmorphic features. Abbreviations: ATA = American Thyroid Association; MTC = medullary thyroid carcinoma; PCC = pheocromocytoma; PHPT = primary hyperparathyroidism; MOD = moderate; H = high; HST = highest.

**Table 2 cancers-12-03268-t002:** Clinical characteristics of MTC patients according to test result.

Feature	*RET* VARIANT DETECTED				*p* Value		
	NONE	ALL	MOD	HST/H	(NONE vs. ALL)	(NONE vs. MOD)	(MOD vs. HST/H)
**Age at diagnosis of MTC (mean)**	56.42	44.45	51.78	11.50	0.010	0.281	<0.001
**Sex, n. Female/tot (%)**	32/48 (66.7)	13/22 (59.1)	10/18 (55.6)	3/4 (75.0)	0.597	0.408	0.616
**Presence of other tumors, n./tot (%)**	14/48 (29.2)	6/22 (27.3)	4/18 (22.2)	2/4 (50.0)	1.000	0.759	0.292
**Positive family history, n./tot (%)**	1/48 (2.1)	15/22 (68.2)	13/18 (72.2)	2/4 (50.0)	<0.001	<0.001	0.565
**Stage, n. T > 1/tot^a^ (%)**	8/22 (36.4)	3/7 (42.9)	2/6 (33.3)	1/1 (100.0)	1.000	1.000	0.429

^a^ Information on stage was available for 29 patients.

**Table 3 cancers-12-03268-t003:** Frequency and predictions of unclassified variants identified.

Nucleotide Variant	Aminoacidic Change	Allele Frequency ^a^	Mean Conservation Score ^b^	Clinvar Class	Varsome Class (Computational Verdicts)	N. Carriers (n. Affected by MTC)
c.2711C > T	p.Ser904Phe	NP	5.5799	LP	LP (10 D vs. 1 B)	8 (5)
c.2372A > T	p.Tyr791Phe	0.00209	5.34	CI	LB (8 D vs. 3 B)	1 (1)
c.2129A > G	p.Lys710Arg	0.00000816	3.72	US	US (8 D vs. 3 B)	1 (0)
c.1946C > T	p.Ser649Leu	0.0003164	4.34	CI	LP (11 D vs. 0 B)	1 (1)
c.1893_1898delCGAGCT	p.Asp631_Leu633delinsGlu	NP	1.2967	NP	LP (1 D vs. 0 B)	1 (1)

^a^ From gnomAD exomes; ^b^ From GERP (Genomic Evolutionary Rate Profiling http://mendel.stanford.edu/SidowLab/downloads/gerp/). Abbreviations: P = pathogenic; LP = likely pathogenic; NP = not present; D = deleterious; B = benign; LB = likely benign; CI = conflicting interpretations; US = uncertain significance.

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
