# Peer review of "Results and Clinical Interpretation of Germline RET Analysis in a Series of Patients with Medullary Thyroid Carcinoma: The Challenge of the Variants of Uncertain Significance"

_cancers, 2020, doi:10.3390/cancers12113268_

Round 1
Reviewer 1 Report
The paper deals with determining the medullary thyroid carcinoma’s risk based on single germline RET mutations and according to the ATA guidelines, both with complete gene analysis and targeted analysis, with a focus on those variants with counselling and management complexity, such as variants of uncertain significance, and rare or novel RET mutations.
The analyses were carried out on 163 cases from 2002 to 2020 and made it possible to detect a novel variant with HST/H risk level and to support the already proposed hypothesis of a MOD risk level for another rare mutation.
One of the most intriguing literature topic regards the possible relationship between c-cell hyperplasia and RET mutations.
Did the authors approach this issue in their series?
Issues:
-
Lines 60-63: more information about the specific risk classes’ therapeutic implications would be needed.
-
Lines: 80-81: miscalculation? 15+33 is 48, not 49.
-
Are RET mutated patients 48, 49 or 50? Because in lines 34 and 80 it’s said they are 49, but by adding 2 cases with HST risk (lines 97-98), 2 with H risk (line 99), 34 with MOD risk (line 99) and 12 with VUS (line 116) the result is 50.
-
Line 223: missing the absolute number of patients under 20 yo.
Author Response
Dear Reviewer,
thank you for your meaningful comments. We have tried to address all the issues you raised. Please find below our point-by-point responses.
The paper deals with determining the medullary thyroid carcinoma’s risk based on single germline RET mutations and according to the ATA guidelines, both with complete gene analysis and targeted analysis, with a focus on those variants with counselling and management complexity, such as variants of uncertain significance, and rare or novel RET mutations.
The analyses were carried out on 163 cases from 2002 to 2020 and made it possible to detect a novel variant with HST/H risk level and to support the already proposed hypothesis of a MOD risk level for another rare mutation.
One of the most intriguing literature topic regards the possible relationship between c-cell hyperplasia and RET mutations.
Did the authors approach this issue in their series?
We thank Reviewer 1 for the important suggestion, as a result of which we have added a brief description of the fact that CCH is a precursor condition for MTC in the first part of the introduction, we have added in the results section our data on the finding of RET variants in patients with CCH and we have added in the discussion the description of our only case of CCH with a RET variant and the clinical implications for this patient.
We have therefore added the following sentence in the first paragraph of the introduction and adjusted the bibliography accordingly: pages 1-2, lines 43-46, “and originates from the parafollicular cells of the thyroid or C-cells, physiologically responsible for the secretion of calcitonin; C-cells hyperplasia (CCH) is considered as the first stage of histological progression that evolves into MTC [3, 4].”.
In Results section, “2.1. Clinical characteristics and genetic test results”, we have added our data: page 2, lines 93-94, “Finally, considering only patients with CCH, 1/3 (33.3%) carried a variant of RET”.
Finally, in the Discussion, we have added the following paragraph: page 6, lines 224-229, “Among patients with CCH, 1/3 (33.3%) carried a RET variant (of MOD risk level): actually she was an asymptomatic patient who was found to have inherited the familial variant and subsequently undertook the surveillance program that led to the diagnosis of CCH. On one hand, this may have represented a successful instance of early diagnosis, provided that CCH is a precursor condition for MTC, on the other hand, we cannot be sure that this benign finding would eventually evolve in a malignant condition and this may have been, conversely, a case of overdiagnosis and overtreatment.”.
Moreover, to make the reading more fluent, throughout the manuscript we have replaced "C-cells hyperplasia" with the abbreviation "CCH".
Issues
- Lines 60-63: more information about the specific risk classes’ therapeutic implications would be needed.
We thank the Reviewer 1 for these suggestions, after which we have deepened the description of these aspects.
We have therefore replaced these lines with the following paragraph: page 2, lines 62-70, “The ATA’s recommended management consists of prophylactic thyroidectomy to be performed as soon as possible in carriers of HST and H risk level variants (within the first year of life and within 5 years, respectively), while for carriers of MOD risk level variants it is suggested to perform annual serum calcitonin screening, and prophylactic thyroidectomy should be performed when values becomes elevated. Screening for PCC (annual dosing of plasma and urinary catecholamines and metabolites) should start at 11 years for carriers of H/HST variants and at 16 years for carriers of MOD variants. For carriers of H and MOD variants, biochemical screening for PHPT is also recommended (annual dosing of serum calcium and parathyroid hormone), starting at 8 and 20 years, respectively [21, 22, 23].”.
- Lines: 80-81: miscalculation? 15+33 is 48, not 49
It was a typographical error, we corrected the number of positives among the subjects subjected to targeted analysis (from 33 to 34) and we therefore modified the percentage accordingly (from 54.1% to 55.7%).
As suggested by Reviewer 2, we have rewritten this whole part and replaced it with the following: pages 2-3, lines 85-100, “Among subjects submitted to RET analysis in our laboratory, 117/163 (71.8%) had personal and/or family history of MTC (114) or CCH (3), while the remaining had personal and/or family history of other pathologies possibly related to RET alterations (PCC, PHPT, intestinal ganglioneuromatosis). The list of cases included in the study is reported in supplementary table 1.
A germline RET variant was identified in 49/163 (30.1%) subjects: 15/102 (14.7%) index cases, who underwent a complete gene analysis, and 34/61 (55.7%) relatives of a RET carrier, who underwent a targeted search for a known variant. Considering only the 72 patients affected by MTC/ CCH, a germline RET variant was identified in 23 (31.9%): 15/16 (93.8%) of those having positive family history and 8/56 (14.3%) of sporadic cases. Finally, considering only patients with CCH, 1/3 (33.3%) carried a variant of RET.
Mean age at MTC diagnosis was 44,5 years in patients with germline RET variants (pathogenetic or considered likely pathogenetic) and 56,42 years in patients with no variants detected (p=0.010). Among patients with known cancer staging, 42.9% of those with RET variants and 36.4% of those with negative analysis had locally advanced disease (T>1). All four MTC patients who also displayed other manifestations of MEN2, were found to carry a pathogenic germline RET variant.”.
- Are RET mutated patients 48, 49 or 50? Because in lines 34 and 80 it’s said they are 49, but by adding 2 cases with HST risk (lines 97-98), 2 with H risk (line 99), 34 with MOD risk (line 99) and 12 with VUS (line 116) the result is 50.
Patients found carrying at least one RET variant are 49, because two variants (a MOD variant and a VUS) were found in the same patient, as described in the paragraph between lines 168 and 176 on page 5.
- Line 223: missing the absolute number of patients under 20 yo
We have replaced the phrase "all those" with the absolute number of patients under 20 yo.
Page 7, line 237, “and 2/2 of those under 20 years of age (100.0%)”.
Reviewer 2 Report
The authors of this study shared with their own sufficient experiences with which germline RET genetic testing was performed in their laboratory of one institution for a long time. The study focused on the clinical role of the variants of uncertain significance which were not yet clarified till recent publications. The manuscript is well described, except for the former part of the result section, which is difficult to follow because of many scattered numbers. Several points should be clarified before publication.
Introduction
- Not bad and fairly described the importance of RET gene testing in medullary thyroid carcinoma and its related conditions like pheochromocytoma/paraganglioma
Results
2.1 clinical characteristics and genetic test results
- Page 2, line 77~ line 91: This part is complex and confusing, very difficult to read and follow. How about using pie charts or other figures to show your data more clearly? Anyways, this part should be more clarified for readers to understand.
2.2 RET test results and clinical correlations
In table 2.
- This table is confusing. I cannot understand why in this table each parameter (Row headers like Sex, Stage, Other tumors, and Family history) has two rows. You don’t need to have ‘M’ row in Sex, ‘T<1’ row in Stage, ‘N’ row in Other tumors, and ‘N’ row in Family history. Only one row per each parameter is enough to present data and statistical significance
- In the ‘Age at diagnosis of MTC (mean)’ row, the ‘NONE vs ALL’ cell is empty. Is it p=0.010?
- Why the total number of ‘Stage’ is different from that of other parameters? (48 vs. 22 in None, 22 vs. 7 in ALL, 18 vs. 6 in MOD, and so on…) This point should be clarified and described.
2.3 Families with unclassified variants
- Well described and OK
Discussion
- Well described and OK
Materials and Methods
4.3 RET analysis
- Data obtained by Sanger sequencing from 2002 to June 2019 and data obtained by NGS since July 2019 can be different in that NGS can get more information than Sanger sequencing and Sanger sequencing might miss several variants, even though RET mutation mostly happens in hot spot area of well-known exons. How many patients were analyzed by NGS data? NGS data for a short time seem to be limited to a small number of patients. This point should be described and discussed.
Author Response
Dear reviewer,
thank you for your consideration to our work. We have changed the manuscript according to your comments. Please find below our point-by-point responses.
The authors of this study shared with their own sufficient experiences with which germline RET genetic testing was performed in their laboratory of one institution for a long time. The study focused on the clinical role of the variants of uncertain significance which were not yet clarified till recent publications. The manuscript is well described, except for the former part of the result section, which is difficult to follow because of many scattered numbers. Several points should be clarified before publication.
Results
2.1 clinical characteristics and genetic test results
- Page 2, line 77~ line 91: This part is complex and confusing, very difficult to read and follow. How about using pie charts or other figures to show your data more clearly? Anyways, this part should be more clarified for readers to understand.
We have rewritten this part trying to make it easier to understand. We think it is now easier to follow even without using summary graphics.
We have therefore replaced this part with the following part: page 2-3, lines 85-100, “Among subjects submitted to RET analysis in our laboratory, 117/163 (71.8%) had personal and/or family history of MTC (114) or CCH (3), while the remaining had personal and/or family history of other pathologies possibly related to RET alterations (PCC, PHPT, intestinal ganglioneuromatosis). The list of cases included in the study is reported in supplementary table 1.
A germline RET variant was identified in 49/163 (30.1%) subjects: 15/102 (14.7%) index cases, who underwent a complete gene analysis, and 34/61 (55.7%) relatives of a RET carrier, who underwent a targeted search for a known variant. Considering only the 72 patients affected by MTC/ CCH, a germline RET variant was identified in 23 (31.9%): 15/16 (93.8%) of those having positive family history and 8/56 (14.3%) of sporadic cases. Finally, considering only patients with CCH, 1/3 (33.3%) carried a variant of RET.
Mean age at MTC diagnosis was 44,5 years in patients with germline RET variants (pathogenetic or considered likely pathogenetic) and 56,42 years in patients with no variants detected (p=0.010). Among patients with known cancer staging, 42.9% of those with RET variants and 36.4% of those with negative analysis had locally advanced disease (T>1). All four MTC patients who also displayed other manifestations of MEN2, were found to carry a pathogenic germline RET variant.”.
2.2 RET test results and clinical correlations
In table 2.
- This table is confusing. I cannot understand why in this table each parameter (Row headers like Sex, Stage, Other tumors, and Family history) has two rows. You don’t need to have ‘M’ row in Sex, ‘T<1’ row in Stage, ‘N’ row in Other tumors, and ‘N’ row in Family history. Only one row per each parameter is enough to present data and statistical significance
- In the ‘Age at diagnosis of MTC (mean)’ row, the ‘NONE vs ALL’ cell is empty. Is it p=0.010?
- Why the total number of ‘Stage’ is different from that of other parameters? (48 vs. 22 in None, 22 vs. 7 in ALL, 18 vs. 6 in MOD, and so on…) This point should be clarified and described.
We have modified the table as suggested by eliminating superfluous rows and inserting the data in the "NONE vs ALL" cell (which was deleted during the review of the editor). Pages 3-4.
Total number of “Stage” is different from the other parameters because this parameter was not known for all patients. We have explained this by adding a footnote at Table 2, page 4, line 129.
Since “Stage” was the only parameter calculated on numbers different from the others, we moved it to the last row of the table.
Materials and Methods
4.3 RET analysis
- Data obtained by Sanger sequencing from 2002 to June 2019 and data obtained by NGS since July 2019 can be different in that NGS can get more information than Sanger sequencing and Sanger sequencing might miss several variants, even though RET mutation mostly happens in hot spot area of well-known exons. How many patients were analyzed by NGS data? NGS data for a short time seem to be limited to a small number of patients. This point should be described and discussed.
This is an important topic of discussion. We have therefore added at the end of this section the number of patients who have performed the genetic analysis with one or the other method: page 8, line 322“Overall, 157 patients underwent Sanger sequencing analysis and 6 the NGS panel analysis”.
Given that to date the NGS analysis has been performed only in a small percentage of patients, we do not have sufficient data to highlight any differences, but we can affirm that Sanger analysis in our case series has shown a great detection rate, probably because of the presence of mutational hotspots in RET gene. We have discussed these aspects at the end of the discussion: pages 7-8, lines 278-284, “One limitation of this study is that most patients underwent the analysis of selected RET exons through Sanger sequencing, which is expected to be less sensitive if compared to whole gene NGS-based analysis. However, since all the variants identified were found using Sanger method, and the percentage of individuals is in line with data previously reported in the literature (even slightly higher for sporadic cases), we can conclude that this testing approach demonstrated satisfactory accuracy in finding RET variants, supporting the evidence that most clinically relevant variant reside in known mutational hotspots [54]”.
Round 2
Reviewer 1 Report
The authors have assessed the issues clarifying doubts and appropriately modifying the paper, in particular about the RET-CCH relationship and the therapeutic implications of the specific risk classes, therefore in my opinion the article can be submitted for publication.
Reviewer 2 Report
Thank you for your efforts.